# Anticandidal Potential of Stem Bark Extract from *Schima superba* and the Identification of Its Major Anticandidal Compound

**DOI:** 10.3390/molecules24081587

**Published:** 2019-04-22

**Authors:** Chun Wu, Hong-Tan Wu, Qing Wang, Guey-Horng Wang, Xue Yi, Yu-Pei Chen, Guang-Xiong Zhou

**Affiliations:** 1Xiamen Key Laboratory of Traditional Chinese Medicine Bio-engineering, Xiamen Medical College, Xiamen 361023, China; ksdchun416@126.com (C.W.); xmwq1122@xmmc.edu.cn (Q.W.); 2Key Laboratory of Functional and Clinical Translational Medicine, Fujian Province University, Xiamen Medical College, Xiamen 361023, China; yixue8866@163.com; 3Department of Medical Technology, Xiamen Medical College, Xiamen 361023, China; 201500010324@xmmc.edu.cn (H.-T.W.); wgh@xmmc.edu.cn (G.-H.W.); 4Application Technique Engineering Center of Natural Cosmeceuticals, College of Fujian Province, Xiamen Medical College, Xiamen 361023, China; 5Guangdong Province Key Laboratory of Pharmacodynamic Constituents of TCM and New Drugs Research, College of Pharmacy, Jinan University, Guangzhou 510632, China

**Keywords:** *Schima superba*, *Candida albicans*, saponins, RNA-seq, sasanquasaponin III

## Abstract

Plant-derived extracts are a promising source of new drugs. *Schima superba* is traditionally used in China for heat clearing, detoxification, and treatment of furuncles. In this study, the anticandidal properties and mechanism of action of *S. superba* (SSE) were explored using a stem bark extract. SSE possessed high polyphenol and saponin contents of 256.6 ± 5.1 and 357.8 ± 31.5 µg/mg, respectively. A clear inhibition zone was observed for *C. albicans* growth through the disc diffusion method and the 50% inhibition of *C. albicans* by SSE was 415.2 µg/mL. Transcriptomic analysis in *C. albicans* treated with different doses of SSE was conducted through RNA-seq. Average values of 6068 genes and 20,842,500 clean reads were identified from each sample. Among these samples, 1680 and 1956 genes were differentially expressed genes (DEGs) from the SSE treatments of 0.2 and 0.4 mg/mL, respectively. *C. albicans* growth was inhibited by the changes in gene expression associated with the cell wall and membrane composition including the regulation of chitin degradation and ergosterol biosynthesis. This result could be reflected in the irregularly wrinkled morphology of the ruptured cell as revealed through SEM analysis. ESI-MS and NMR analyses revealed that the major compound purified from SSE was sasanquasaponin III and the 50% inhibition of *C. albicans* was 93.1 µg/mL. In summary, the traditional Chinese medicine *S. superba* can be applied as an anticandidal agent in complementary and alternative medicine.

## 1. Introduction

*Schima superba* Gardn. et Champ. (Theaceae) is a dominant broad-leaved evergreen tree widely distributed in the subtropical regions of China. This traditional Chinese medicine is used for heat clearing, detoxification, and furuncle treatment; it is also used as an insecticide agent [1]. The root and stem bark of *S. superba* can dispel wind dampness, relieve pain, serve as an emetic, and induce diuresis for removing edema [2]. Bioactive compounds from *S. superba* have been widely explored. An abundance of phenolics in *S. superba* with antimicrobial and antioxidant potential was noted [3,4]. Some saponins from *S. superba* leaves have antifungal effects against *Magnaporthe oryzae*, which can cause a highly damaging disease in rice [5,6]. They were identified as two oleanane-type triterpenoid saponins, schimasuoside A (1) and schimasuoside B (2) [6]. Two compounds, namely, (7*R*,8*S*)-1-(3,4-dimethoxyphenyl)-2-*O*-(2-methoxy-4-ω-hydroxypropylphenyl)propane-1,3-diol and threo-1-(4-hydroxy-3-methoxy- phenyl)-1-methoxy-2-{4-[1-formyl-(*E*)-vinyl]-2-methoxyphenoxy}-3-propanol, characterized from the stems of *S. superba*, have inhibitory activities against some human cancer cell lines [7]. Eight oleanane-type triterpenoid saponins, that is, schisusaponins A–H, have also been identified from the root bark of *S. superba*; among them, schisusaponins C and E exhibit cytotoxicity on the B16 melanoma cell line with IC_50_ values of 10.08 and 10.89 mM, respectively [8]. 

*Candida albicans* is a major opportunistic fungal pathogen in humans that can induce superficial and severe infections in individuals or immunocompromised hosts [9]. Well-known antifungal agents, such as the azoles, including imidazoles and triazoles, are generally utilized to treat infections [10]. They inhibit lanosterol 14-*α*-demethylase in the biosynthesis of ergosterol, a major component of fungal membrane [11]. However, drug-resistant isolates have emerged because of the widespread overuse of azole drugs [12]. Hence, the antimicrobial activity of plant-derived substances, which can be developed as sources of new drugs, has been studied [13]. Anticandidal susceptibility is exhibited by various extracts derived from different plants, such as *Anagallis arvensis* L. (Primulaceae), *Sideroxylon obtusifolium* T. D. Penn (Sapotacea), *Syzygium cumini* (L.) Skeels (Myrtaceae), *Combretum zeyheri*, *Hosta plantaginea* (Lam.) Aschers, and ornamental tobacco [14,15,16,17,18]. The extracts with potent activity against *C. albicans* include saponins such as anagallisin C, hostaside I, and hostaside II, and peptides, such as NaD1 defensin. 

In this study, an extract from the stem bark of *S. superba,* designated as *S. superba* extract (SSE) was used to evaluate the antimicrobial activity. SSE was subjected to anticandidal susceptibility testing, and RNA-seq analysis of *C. albicans* at different SSE doses was conducted for molecular examination. The morphological characteristics of *C. albicans* were detected through scanning electron microscopy (SEM) to investigate the effect of SSE. In addition, the component of SSE was identified and quantified through ultraperformance liquid chromatography (UPLC), electrospray ionization mass spectrometry (ESI-MS), and nuclear magnetic resonance (NMR). Sasanquasaponin III purified from SSE was also evaluated for its anticandidal ability. 

## 2. Results

### 2.1. Antioxidant Activity of SSE

The antioxidant activity of SSE was analyzed through DPPH and ABTS+ radical scavenging assays. SSE exhibited a high antioxidant activity (Figure 1) and scavenged DPPH and ABTS+ radical and the positive control of ascorbic acid (purity > 99%) in a dose-dependent manner. Regression analysis revealed that the inhibition rate of DPPH radicals was 50% at 0.11 mg/mL SSE. When the SSE concentration was increased to 0.25 mg/mL, the inhibition of DPPH radicals was over 80%. A similar result was obtained from the ABTS+ radical scavenging assay. The addition of 0.0625 mg/mL SSE significantly arrested ABTS+ to 64.7%. More than 90% scavenging rate of ABTS+ was achieved with 0.5 mg/mL SSE treatment. 

### 2.2. Antimicrobial Assay of SSE

The response of four bacterial and yeast pathogens was determined through the disc diffusion method to comprehensively detect the antimicrobial ability of SSE. No evident clear zone was observed in the bacterial pathogens, including *Escherichia coli*, *Klebsiella oxytoca*, and *Staphylococcus aureus* (data not shown). Conversely, *C. albicans* was susceptible to SSE, and an clear antifungal zone could be observed when 0.0625 mg of SSE was added (Figure 2). Different SSE concentrations (0, 0.03125, 0.0625, 0.125, 0.25, 0.5, 1, and 2 mg/mL) were added to the liquid culture of *C. albicans* at 37 °C for 24 h of cultivation to detect the half maximal inhibitory concentration of SSE. SSE obviously inhibited *C. albicans* growth in a dose-dependent manner (Figure 3). A 74.1% inhibition rate of *C. albicans* was attained with 0.5 mg/mL SSE treatment. The 50% inhibition of *C. albicans* by SSE was calculated at 415.2 µg/mL SSE.

### 2.3. Total Polyphenol and Saponin Determination of SSE

The total polyphenol content of SSE was determined using Folin–Ciocalteu method. SSE contained 256.6 ± 5.1 µg/mg corresponding to gallic acid (purity > 98%) according to the absorbance measured at 760 nm. The total saponin content of SSE was measured by determining the absorbance at 548 nm. The result showed that SSE contained 357.8 ± 31.5 µg/mg corresponding to oleanic acid (purity > 98%).

### 2.4. RNA-seq Analysis of C. Albicans with SSE Treatment

*C. albicans* was treated with SSE at 0, 0.2, and 0.4 mg/mL for RNA-seq analysis. An average of 20,842,500 clean reads was obtained from each sample with duplication. After quality filtering and trimming, an average of 92.91% clean reads was mapped to the reference genome of *C. albicans* SC5314, indicating that the samples were comparable. Pearson correlation coefficients were determined to reflect the gene expression correlation between the two samples. The duplication of the same treatment had a high correlation with an average of 0.97. The RNA-seq reads were deposited in the NCBI SRA under the accession numbers BioProject PRJNA475454 and SRA SRP150296.

The RNA sequencing results showed that an average of 6068 genes was detected in these samples. The volcano plots of the DEGs revealed that 1680 and 1956 genes in *C. albicans* were differentially expressed after receiving 0.2 and 0.4 mg/mL SSE, respectively, and either an increase or decrease of more than twofold change was observed (Figure 4). The amount of downregulated genes increased when the SSE concentration was increased. Similar results of 252 downregulated genes and 63 upregulated genes were obtained by comparing 0.2 and 0.4 mg/mL SSE treatment, respectively.

The 15 most upregulated and downregulated genes by the SSE treatment in a dose-dependent manner are presented in Appendix A. The downregulated genes have various molecular functions, such as involvement in oxidoreductase activity, monooxygenase activity, membrane channel activity, transporter activity, metal ion, nicotinamide adenine dinucleotide phosphate (NADP), adenosine triphosphate (ATP), and heme binding. By contrast, the 15 most upregulated genes are associated with major facilitator superfamily (MFS) transporter, superoxide dismutase (SOD), phenylpyruvate decarboxylase, chitinase, NADPH dehydrogenase, methylglyoxal catabolic process, and phosphate transporter.

Well-known antifungal drugs, such as azoles, can regulate *C. albicans*, and arrest the fungal ergosterol biosynthesis pathway and multidrug transporter genes responding to the resistance of azole antifungal agents [11,19]. Therefore, the effects of SSE on the genes involved in ergosterol biosynthesis pathway and multidrug transporter are shown in Table 1. Another two genes encoding C-5 sterol desaturase and methylsterol monooxygenase with over a twofold change were significantly downregulated in the sterol biosynthesis pathway. Two genes related to the MFS (MDR1 and HOL1) for multidrug resistance protein were upregulated with the increase in SSE treatment. The fungal cell wall is necessary to maintain cell morphology against environmental stress [20]. Nevertheless, upregulated (CHT1 and CHT4) and downregulated (CHT2) the genes involved in chitin degradation were also observed. Eight genes comprising four upregulated and four downregulated genes were selected for real-time RT-PCR to confirm the RNA-seq results. The expression levels obtained through real-time RT-PCR were consistent with the change in RNA-seq in a dose-dependent manner (Appendix A).

### 2.5. SEM of C. Albicans with SSE Treatment

SEM micrographs are shown in Figure 5 to illustrate the effect of SSE against *C. albicans* cells. A typical *C. albicans* cell treated with SSE showed an oval and smooth appearance as compared with the control. Our results also revealed a mucilaginous material that could be a polysaccharide secreted for biofilm synthesis. Nevertheless, after 40 h of SSE exposure on an agar plate with 0.25 mg/mL, the treated cells appeared to be irregularly wrinkled and had completely collapsed. A clear broken hole was also observed in the ruptured cell. 

### 2.6. Saponin Identification from SSE by UPLC, ESI-MS and NMR Analyses

SSE was further utilized to purify the fraction and confirm the bioactive substance working against *C. albicans*. SSE was subjected to UPLC analysis, and a major peak was observed at a retention time of 15.4 min (Figure 6). 

The dominant compound **1** was examined through NMR and ESI-MS analyses. The result indicated the following characteristics of C_59_H_94_O_26_; ${\left[\alpha \right]}_{D}^{25}$-18.1 (*c* 0.30, MeOH), ESI-MS *m/z* 1217.9 [M-H]^−^ (Appendix A), ^1^H-NMR (C_5_D_5_N, 600 MHz) (Appendix A), and ^13^C-NMR (C_5_D_5_N, 125 MHz) (Table 2, Appendix A). The MS and NMR data of the compound were compared with those in previous studies, and sasanquasaponin III was thus identified (Figure 7) [21]. 

Sasanquasaponin III was harvested through TLC and preparative HPLC to verify its anticandidal activity. *C. albicans* was used for determination by disc diffusion method, and different sasanquasaponin III concentrations were used for half maximal inhibitory concentrations. The results revealed that an evident clear zone was found when 0.0625 mg was added (Figure 8). Sasanquasaponin III significantly inhibited *C. albicans* growth in a dose-dependent manner (Figure 9). After 0.25 mg/mL sasanquasaponin III was administered at 37 °C for 24 h, a 94.9% inhibition rate of *C. albicans* was achieved. The 50% inhibition of *C. albicans* by sasanquasaponin III was calculated at 93.1 µg/mL.

## 3. Discussion

Plant-derived extracts have been studied for new drug development. *C. albicans* is susceptible to 142 natural products according to 111 research documents filed between 1969 and 2015 [13]. *S. superba* is widely distributed in southeast China and composed of bioactive ingredients that can be applied in cancer cell lines [7,8]. Therefore, the valuable substances from *S. superba* are worthy of being explored. In the present research, the SSE extract obtained from the stem bark of *S. superba* exhibited antioxidant and anticandidal activities. SSE possessed high polyphenolic compounds corresponding to DPPH and ABTS+ radical scavenging assays (Figure 1) [3]. To the best of our knowledge, the antimicrobial activity of *S. superba* extract has not been investigated. Here, SSE exhibited a dose-dependent anticandidal effect (Figure 2). However, no bacteria, such as bacterial pathogens, including *E. coli*, *K. oxytoca,* and *S. aureus*, were sensitive to SSE, as revealed by the disc diffusion method. The 50% inhibition of *C. albicans* was identified at 415.2 µg/mL SSE (Figure 3). Therefore, SSE could serve as a source of new drugs for *C. albicans* treatment. 

The antifungal effect of SSE may involve multitarget and multichannel actions because SSE has multiple components. The fungal cell wall is composed of two covalently cross-linked polysaccharides, namely, *β*-glucan and chitin, which are responsible for its structural shape [22]. *C. albicans* dynamically changes the composition of the cell wall by breaking and reforming the chemical bonds of polysaccharides in response to environmental stresses [23]. Thus, the gene expression of the cell wall can be altered to reduce the stress in *C. albicans*. In general, the highest chitinase activity is contributed by CHT3 followed by CHT2; conversely, low chitinase activities are associated with CHT1 and CHT4 [24]. In this study, several genes involved in chitin degradation were upregulated and downregulated (Table 1). With regard to osmosis, SSE was hydrophilic and had penetrated *C. albicans* cells and even disturbed its gene expression. Therefore, our results were similar to those observed in micafungin (MFG) treatment, an antifungal drug in which CHT1 and CHT2 are upregulated and downregulated, respectively [24]. By contrast, CHT3 remained unaltered, whereas CHT4 was markedly upregulated after the SSE treatment. However, the low expression levels of CHT2 and CHT3 were consistent with the reports on the resistance to MFG and might be responsible for the fungal tolerance against cell wall stress.

The fungal plasma membrane is composed of a lipid bilayer embedded with proteins, cholesterol, and ergosterol. Lanosterol 14-*α*-demethylase is a critical enzyme involved in ergosterol biosynthesis and therefore is targeted by the widely used antifungal drugs of azoles [11]. Inhibiting lanosterol 14-*α*-demethylase can cause the accumulation of 14-*α*-methylsterols on the fungal surface, thereby changing in the permeability and rigidity of plasma membrane, and halting the fungal growth [25]. However, lanosterol 14-*α*-demethylase was upregulated in the current study (Table 1). Conversely, the downstream genes of ergosterol biosynthesis with C-5 sterol desaturase and methylsterol monooxygenase were significantly downregulated. Thus, SSE could interfere with certain ergosterol biosynthesis genes and reduce the composition of the cell membrane. Meanwhile, several multidrug resistance efflux transporter genes are correlated with the resistance to antifungal drugs [10]. Multidrug resistance-associated proteins, namely, CDR1 and CDR2, are involved in the mechanisms of azole efflux and were not upregulated after 24 h of treating *C. albicans* with itraconazole [10]. In our study, CDR1 was significantly downregulated, whereas the gene expression level of CDR2 was not changed. This result was consistent with that from itraconazole treatment but was completely different from the transcriptomic analysis of *C. albicans* treated with Huanglian Jiedu decoction, an aqueous extract of four herbs, namely, *Coptidis Rhizoma*, *Scutellariae Radix*, *Phellodendri Cortex*, and *Gardeniae Fructus* [26], possibly because of their different antifungal mechanisms.

SSE was sufficient to alter cellular morphology. The damage to the cell wall and the cell membrane was verified by an irregular, wrinkled, and uneven appearance as observed in *C. albicans* treated with 0.25 mg/mL SSE (Figure 5). SSE induced biofilm deformation and breakage of cell–cell connections. This result was similar to previous findings on *C. albicans* treated with the synthetic human *β*-defensin 3-C15 peptide and root extract of *Juglans regia* [27,28]. The fluctuant gene expression involved in chitin degradation and ergosterol biosynthesis was probably responsible for the morphological change in the cell wall and membrane. 

Saponins have a cardioprotective potential because of their antihypoxic, calcium ion regulation/inotropic, anti-atherosclerotic, and hypolipidemic effects, anoxia/reoxygenation, cardiocyte apoptosis, and vasodilatory/cardiac depressant [29]. Additionally, saponins from plants exhibit anticandidal susceptibilities, such as anagallisin C, hostaside I, hostaside II, and CAY1, thus provide a promising evidence for their antifungal ability [14,17,30,31]. In our research, a high saponin content with 357.8 µg/mg oleanic acid was observed in SSE. Therefore, saponin was further purified from SSE, and sasanquasaponin III was identified as the major compound through UPLC analysis (Figure 6 and Figure 7). Sasanquasaponin III, a kind of triterpenoid saponin, was first isolated from the flower buds of *Camellia sasanqua*, which is an ornamental plant in Japan [21]. This substance inhibited the release of *β*-hexosaminidase from rat basophile leukemia cells at 3 µM with 51.6% inhibition rate for antiallergic activity. In addition, sasanquasaponin III can be found in the root bark of *S. superba* [8]. Thus, sasanquasaponin III widely exists in the stem and root of *S. superba*. The fungicidal effect of sasanquasaponin III on *C. albicans* was confirmed in this study, and the 50% inhibition of *C. albicans* was identified at a concentration of 93.1 µg/mL (Figure 8 and Figure 9). In general, saponins can interact with the cell wall and membrane to stimulate cell lysis [30,31,32]. RNA-seq and morphological observation showed that the anticandidal effect of SSE and sasanquasaponin III was consistent with the fungicidal property of saponin. This study was the first to characterize the anticandidal function of sasanquasaponin III.

## 4. Materials and Methods

### 4.1. Plant Material and Extraction

The dried stem bark of *S. superba* was collected from Sanming (longitude: 116.820837, latitude: 26.972907), Fujian, China, in March 2016 and authenticated by Prof. Ming-Jun Wang (Department of Pharmacy, Xiamen Medical College). A voucher specimen (No. MH20160311) was deposited in the Xiamen Key Laboratory of Biotechnology of Traditional Chinese Medicine, Xiamen Medical College. The dried and powdered stem bark of *S. superba* (2.5 kg) was soaked in 70% (*v/v*) ethanol (EtOH) at room temperature thrice (3 × 6 L) for 2 h at each time. The EtOH solution was filtered and concentrated under reduced pressure to yield a crude extract (235.2 g) designated as SSE.

### 4.2. Antioxidant Activity by 2,2-diphenyl-1-picrylhydrazyl (DPPH) Radical Scavenging Assay

A DPPH radical scavenging assay was conducted as described by Bersuder et al. [33]. Different SSE concentrations (0.0625, 0.125, 0.25, 0.5, and 1 mg/mL) were mixed with 0.2 mL of 10 mM DPPH (dissolved in methanol to a final concentration of 2 mM). The mixture was allowed to stand in the dark at room temperature for 30 min. Distilled water was used as a substitute for the SSE sample as the background (absorption control), and ascorbic acid (AA) was set as a positive control. The remaining DPPH radical was analyzed on a plate reader (Molecular Devices, Sunnyvale, CA, USA) at 517 nm.

### 4.3. Antioxidant Activity by ABTS+ Radical Scavenging Assay

An ABTS+ radical scavenging assay was conducted using a total antioxidant capacity assay kit with the ABTS method (Beyotime Institute of Biotechnology, Shanghai, China). An ABTS+ radical solution was prepared by oxidizing the reagents with ABTS+ and oxidation solution with a volume ratio of 1:1. The mixture was allowed to stand in the dark at room temperature overnight. Different SSE concentrations (0.0625, 0.125, 0.25, 0.5, and 1 mg/mL) were mixed with 0.2 mL of ABTS+ radical solution for 10 min. Distilled water was a substitute for the SSE sample as the background (absorption control), and AA was used as a positive control. The remaining ABTS+ radical was analyzed on a plate reader (Molecular Devices) at 620 nm.

### 4.4. Antimicrobial Assay

A pathogen antagonistic dosage assay was conducted through the disc diffusion method [34]. *C. albicans* (ATCC10231) was cultivated in Sabouraud dextrose broth at 37 °C for 2 days. The culture was adjusted to obtain a concentration of approximately 10^8^ CFU/mL. One hundred microliters of culture suspension was placed on the petri dishes of Sabouraud dextrose agar. Then, 6 mm paper discs impregnated with 20 µL of the SSE and sasanquasaponin III at the concentration of 100, 50, 25, 12.5, 6.25, 3.125, 1.5625, and 0 mg/mL dissolved in distilled water to obtain the concentration of 2, 1, 0.5, 0.25, 0.125, 0.0625, 0.03125, and 0 mg/disc, respectively. Different concentrations of SSE and sasanquasaponin III from 0 mg to 2 mg were arranged on the plate with *C. albicans*. The distance between discs was 2 cm. The antibiotic disc (Amphotericin B, ROSCO, Taastrup, Denmark) was used as the positive control. The plates were cultivated at 37 °C, and the clear zone of growth inhibitions was observed. The microbroth dilution was performed in 96-well plate supplemented with Sabouraud dextrose broth [35]. Different concentrations of SSE and sasanquasaponin III (0, 0.0625, 0.125, 0.25, 0.5, 1, and 2 mg/mL) were obtained by a twofold serial dilution in the Sabouraud dextrose broth at a final volume of 50 µL, respectively. The liquid culture of *C. albicans* was adjusted with OD_600_ of approximately 0.2. A microbial suspension (50 µL) was added to each well and the mixture (100 µL) was incubated at 37 °C. Afterward, the 96-well plate treated with SSE and sasanquasaponin III was analyzed and determined with a plate reader at 600 nm.

### 4.5. Detection of Total Polyphenols and Saponins

The content of total polyphenols was measured using the Folin–Ciocalteu method. The SSE sample (0.5 mL) was mixed with 0.5 mL of Folin–Ciocalteu reagent for 5 min. Then, 0.5 mL of 10% sodium carbonate solution was added to the mixture and left for 30 min in the dark. The mixture was centrifuged at 10,000 × *g* for 10 min at room temperature. The supernatant was analyzed and determined with a plate reader at 760 nm. Gallic acid was used as a positive control to calculate a standard curve (y = 0.0296x − 0.0085; R^2^ = 0.977). The content of saponins was determined in accordance with previously described methods [36] with modifications. The SSE sample was dissolved in 0.2 mL of 5% vanillin/glacial acetic acid and 0.8 mL of perchloric acid at 60 °C for 20 min. The mixture was cooled down for 5 min, analyzed, and determined with a plate reader at 548 nm. Oleanic acid was used as a positive control to calculate a standard curve (y = 0.5956x + 0.1045; R^2^ = 0.9987).

### 4.6. RNA-Seq Analysis of C. Albicans

The cultivated *C. albicans* broth was introduced to the flesh Sabouraud dextrose medium, with OD_600_ of approximately 0.2. Different SSE concentrations (0, 0.2, and 0.4 mg/mL) with duplication were added to the broth at 37 °C for 24 h of cultivation. *C. albicans* specimens were then harvested, and total RNA was obtained using a yeast RNA kit (OMEGA Bio-tek, Guangzhou, China). NanoDrop 2000 (ThermoFisher Scientific, Wilmington, DE, USA) was used for RNA quantification. cDNA synthesis, library construction, cDNA sequencing, and cDNA *de novo* analysis were performed by the Beijing Genomics Institute (BGI, Shenzen, China). First-strand cDNA was synthesized through reverse transcription with random hexamers followed by end repair, polymerase chain reaction (PCR) amplification, denaturation, and cyclization in accordance with a BGI experimental workflow [37]. cDNA sequencing was performed on a BGISEQ-500 platform, and cDNA reads were analyzed on the basis of the BGI bioinformatics workflow. Sequencing reads were trimmed and screened with SOAPnuke to acquire clean reads. *C. albicans* SC5314 was used as a reference. Pearson correlation coefficients were determined using cor by the R software. cDNA *de novo* analysis was conducted through hierarchical indexing for spliced alignment of transcripts (HISAT) for genome mapping [38], RSEM for gene expression analysis [39], cluster for gene expression cluster analysis [40,41], DEGseq for differentially expressed gene (DEG) [42], GO annotation for gene ontology analysis [43], KEGG annotation for DEG pathway analysis [44], and BLAST for gene prediction. The RNA-seq data were deposited at DDBJ/ENA/GenBank under the accession numbers BioProject PRJNA475454 and SRA SRP150296.

### 4.7. Quantitation of Gene Expression by Real-Time RT-PCR

First-strand cDNA was synthesized with a HiFi-MMLV cDNA kit with oligo (dT) and random hexamers (CWBiotech, Beijing, China) at 42 °C for 1 h and used as a template for real-time PCR. Real-time PCR was performed using an UltraSYBR Mixture (CWBiotech) in a final volume of 12.5 μL (2 × UltraSYBR Mixture 6.25 μL, 10 μM Forward primer 0.25 μL, 10 μM Reverse primer 0.25 μL, cDNA 2 μL, and RNase-free water 3.75 μL) for each reaction in the Roche LightCycler^®^ 480 System (Roche Group, Switzerland). The thermal cycling parameters consisted of initial heating at 95 °C for 15 min, denaturation at 95 °C for 10 s, and annealing and extension at 58 °C for 1 min with amplification of 40 cycles. The primer sets of real-time PCR were shown in Appendix A. The eight genes comprising four upregulated and four downregulated genes were randomly selected. Gene expression levels were measured thrice, and the mean of these values was used for analysis. The relative gene expression level was calculated with the 2^−ΔΔCT^ method.

### 4.8. SEM Analysis

The cultivated *C. albicans* was spread on a plate containing Sabouraud dextrose agar with or without SSE (0.2 mg/mL) at 37 °C for 40 h cultivation. Then, the agar plate was examined under a field emission scanning electron microscope (FESEM; FEI Quanta 450 FEG FESEM, Hillsboro, OR, USA) equipped with Quorum PP3000T (Quorum Technologies Ltd., Laughton, UK). The agar samples were vitrified using liquid nitrogen slush at −200 °C. The vitrified samples were sublimated at −90 °C for 12 min and then coated with platinum by Quorum PP3000T (Quorum Technologies Ltd.). The sample images were obtained with a FESEM (FEI Quanta 450 FEG FESEM) at an electron beam strength of 15 kV.

### 4.9. Purification and Identification of SSE by UPLC, ESI-MS, and NMR Analyses

SSE was analyzed through UPLC performed on a Waters ACQUITY UPLC HSS T3 column (2.1 mm × 100 mm, 1.8 μm) (Waters, Milford, MA, USA) with a mobile phase consisting of acetonitrile (A) and 0.1% trifluoroacetic acid water (B) at a flow rate of 0.3 mL/min (0–8 min, 12% A → 20% A; 8–15 min, 20% A → 55% A; 15–18 min, 55% A → 86% A). The detection wavelength was 210 nm, the column temperature was controlled at 30 °C, and the injection volume was 2 μL. The major compound could be separated and obtained with an ODS gel column (180 g, 3.5 cm × 40 cm) (50 μm, YMC, Kyoto, Japan) eluted with various H_2_O-MeOH (70:30, 50:50, 30:70, and 0:100, each at 3 L) in gradient to yield 13 fractions (Fr 1–13) based on their TLC patterns. Fr 10 (30.3 g) was subjected to a silica gel column (200–300 mesh, Qingdao Marine Chemical Inc., Qingdao, China) and eluted with CH_2_Cl_2_-MeOH (100:0, 98:2, 95:5, 9:1, 8:2, 7:3, 6:4, 5:5, and 0:100) in gradient to yield 20 fractions (Sfr 1–20) based on their TLC patterns. Sfr18 (365 mg) was purified through preparative HPLC (Shimadzu chromatograph LC-6AD, Osaka, Japan) by using a reversed-phase C_18_ column (5 μm, 20 mm × 250 mm; Cosmosil, Kyoto, Japan) and 45% CH_3_CN-H_2_O (7 mL/min) to produce compound **1** (52.6 mg, *t*_R_ 25.5 min). Optical rotations were performed using a JASCO P-1030 automatic digital polarimeter. NMR spectra were obtained on a Bruker AV-600 spectrometer with TMS (Bruker, CA, USA) as an internal standard, and chemical shifts were expressed in *δ* (ppm). ESI-MS data were determined on a Finnigan LCQ Advantage Max mass spectrometer (Thermo Electron, Lowell, MA, USA). Compound **1** was identified as sasanquasaponin III through ESI-MS and NMR.

## 5. Conclusions

Our results demonstrated the anticandidal capacity of the stem bark extract from *S. superba*. RNA-seq and SEM results suggested that SSE could interfere in the cell wall and cell membrane composition, resulting in an irregular twisting and death of cells. Further purification and identification revealed that the major anticandidal compound was sasanquasaponin III. These findings indicated that *S. superba* might provide an alternative to anticandidal treatment.

## Figures and Tables

**Figure 1 molecules-24-01587-f001:**
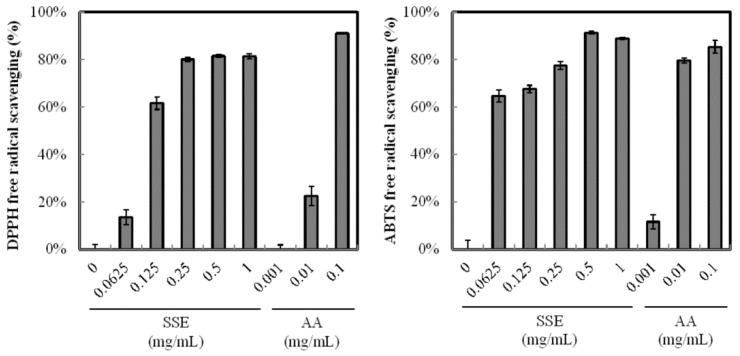
Antioxidant activity of SSE. Radical scavenging capability was measured by the DPPH and ABTS+ method. Ascorbic acid (AA) was used as positive control. Results are mean ± S.D. (*n* = 3).

**Figure 2 molecules-24-01587-f002:**
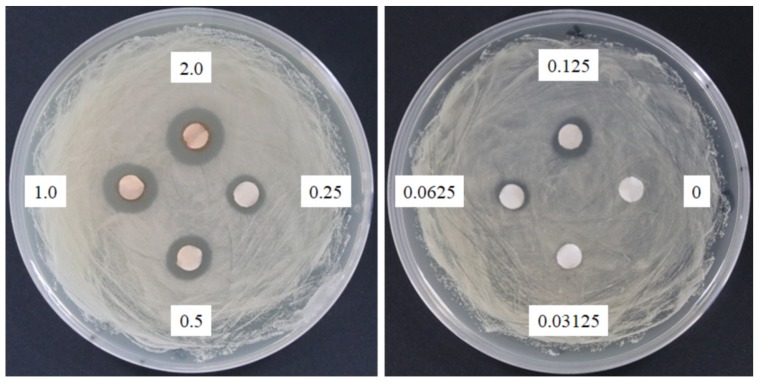
Anticandidal activity analysis of SSE. The 6 mm paper discs with different concentrations of SSE (0, 0.03125, 0.0625, 0.125, 0.25, 0.5, 1, and 2 mg) were placed on the plates with spread *C. albicans*. The plates were cultured at 37 °C for 24 h, and the growth inhibitions of *C. albicans* were observed.

**Figure 3 molecules-24-01587-f003:**
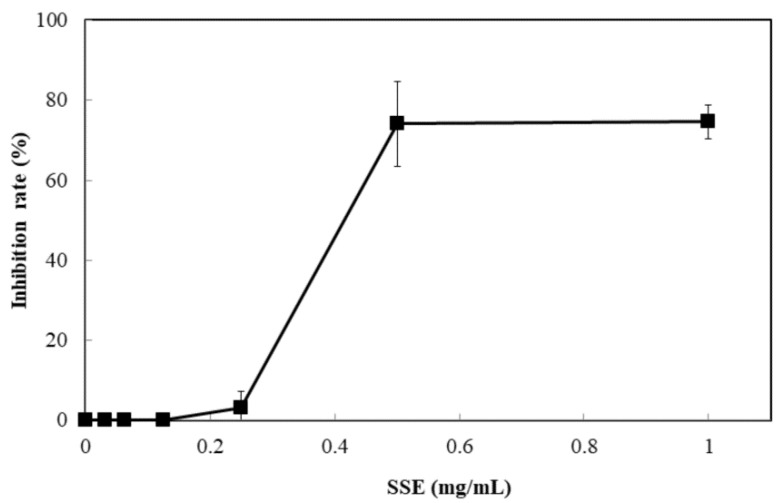
Anticandidal activity analysis of SSE. The liquid culture of *C. albicans* was introduced into 96-well plate and different concentrations of SSE (0, 0.03125, 0.0625, 0.125, 0.25, 0.5 and 1 mg/mL) were added. The 96-well plate treated with SSE was cultivated at 37 °C for 24 h and determined by a plate reader at 600 nm. Results are mean ± S.D. (*n* = 3).

**Figure 4 molecules-24-01587-f004:**
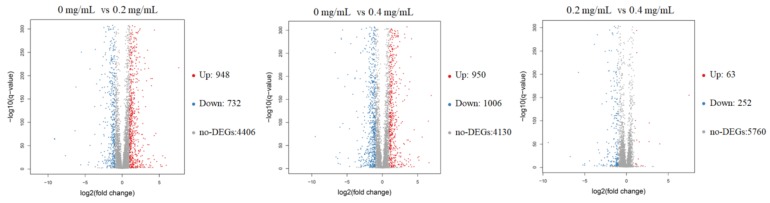
The volcano plots of the differentially expressed genes of *C. albicans* treated with SSE at the concentrations of 0, 0.2, and 0.4 mg/mL for RNA-seq analysis. The red and blue dots indicated the significantly differentially expressed genes with up-regulation and down-regulation, respectively. The red dot indicates up-regulation (Up); the blue dot indicates down-regulation (Down); the grey dot indicates no differentially expressed genes (no-DEGs).

**Figure 5 molecules-24-01587-f005:**
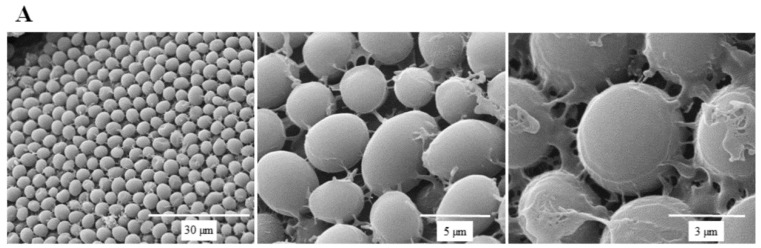
Scanning electron micrographs of *C. albicans*. (**A**) A scanning electron micrograph of *C. albicans* cells after 40 h cultivation at 37 °C. (**B**) A scanning electron micrograph of *C. albicans* cells after 40 h of SSE exposure on an agar plate with 0.25 mg/mL at 37 °C.

**Figure 6 molecules-24-01587-f006:**
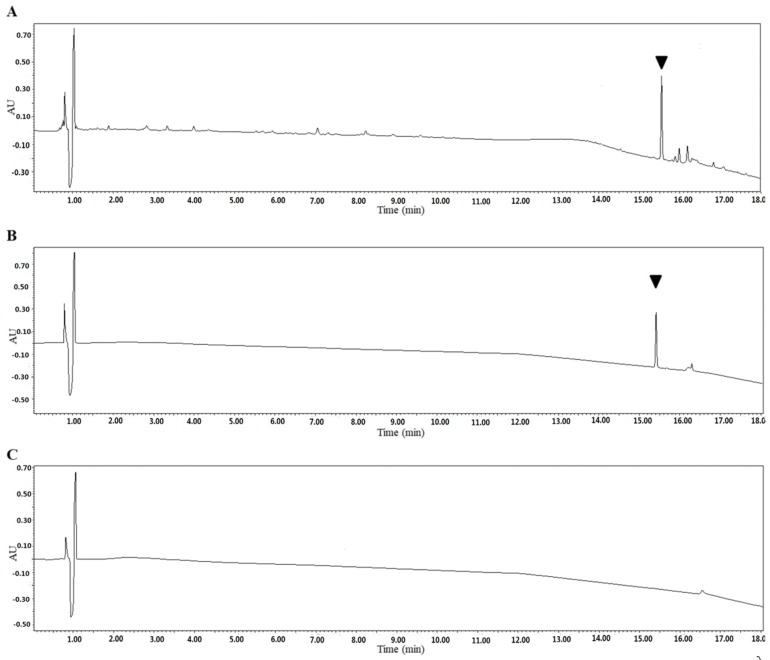
Identification of the SSE by UPLC. UV light absorption spectrum of the SSE at the wavelength of 210 nm. (**A**) UPLC profile of the SSE. (**B**) UPLC profile of the major compound after purification. (**C**) UPLC profile of the control without the SSE. The arrow indicated the major compound.

**Figure 7 molecules-24-01587-f007:**
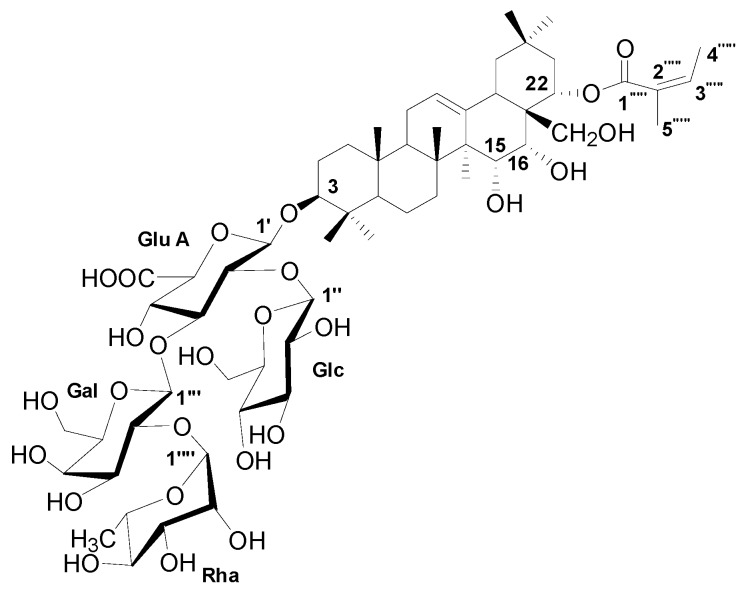
The chemical structure of compound **1**.

**Figure 8 molecules-24-01587-f008:**
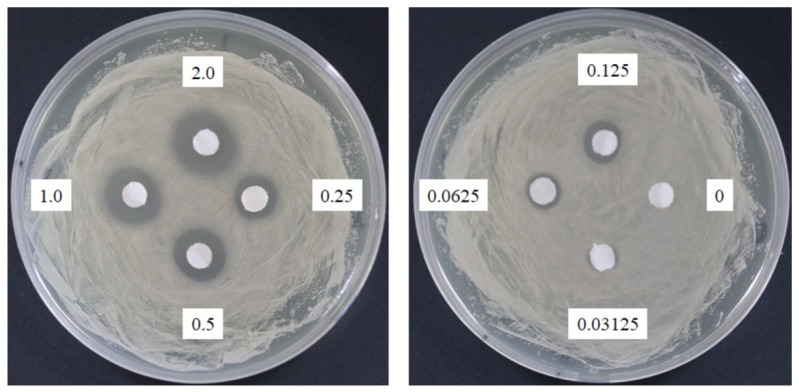
Anticandidal activity analysis of sasanquasaponin III. The 6 mm paper discs with different concentrations of sasanquasaponin III (0, 0.03125, 0.0625, 0.125, 0.25, 0.5, 1, and 2 mg) were placed on the plates with spread *C. albicans*. The plates were cultured at 37 °C for 24 h, and the growth inhibitions of *C. albicans* were observed.

**Figure 9 molecules-24-01587-f009:**
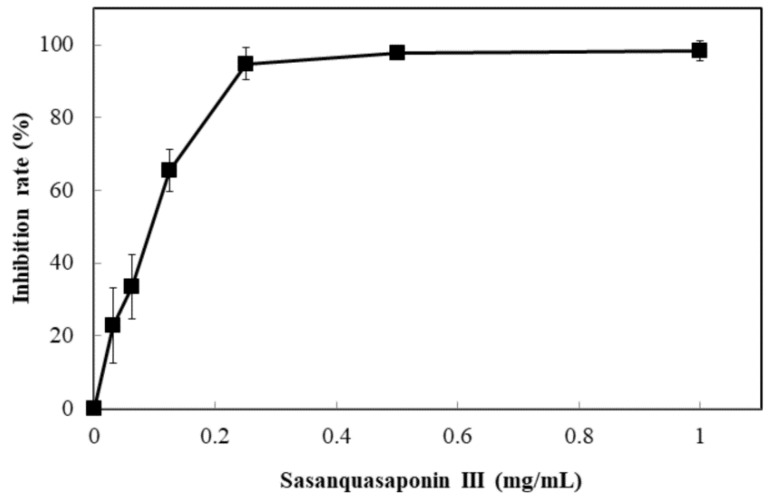
Anticandidal activity analysis of sasanquasaponin III. The liquid culture of *C. albicans* was introduced into 96-well plate and different concentrations of SSE (0, 0.03125, 0.0625, 0.125, 0.25, 0.5, and 1 mg/mL) was added. The 96-well plate treated with sasanquasaponin III was cultivated at 37 °C for 24 h and determined by a plate reader at 600 nm. Results are mean ± S.D. (*n* = 3).

**Table 1 molecules-24-01587-t001:** Response to SSE of the genes involved in ergosterol biosynthesis, multidrug resistance and chitin degradation of *C. albicans* in a dose-dependent manner.

Function	ID in GenBank	Protein	Annotation	Size (aa)	log_2_ Ratio	
					0.2 mg/mL	0.4 mg/mL
Ergosterol biosynthesis	XP_711894.1	ERG1	squalene monooxygenase	496	−0.30	−0.12
	XP_713577.1	ERG3	C-5 sterol desaturase	386	−1.72	−1.84
	XP_722612.2	ERG7	Lanosterol synthase	728	−0.42	−0.73
	XP_722678.1	ERG8	Phosphomevalonate kinase	432	0.93	1.18
	XP_714460.2	ERG9	Bifunctional farnesyl-diphosphate farnesyltransferase/squalene synthase	448	0.26	0.34
	XP_716761.1	ERG11	Lanosterol 14α-demethylase	528	0.43	0.89
	XP_722703.1	ERG251	Methylsterol monooxygenase	321	−0.98	−1.14
	XP_715564.1	ERG26	Sterol-4-alpha-carboxylate 3-dehydrogenase	350	0.20	0.14
	XP_717931.1	ERG27	3-Ketosteroid reductase	346	1.22	1.17
Multidrug resistance	XP_712090.2	ATM1	ATP-binding cassette Fe/S cluster precursor transporter	750	−0.46	−0.64
	XP_718280.1	MDL1	ATP-binding cassette permease	685	−0.16	−0.85
	XP_717637.1	MLT1	ATP-binding cassette	1606	0.44	0.96
	XP_719165.2	MDR1	Multidrug resistance protein: plasma membrane MDR/MFS multidrug efflux pump	564	1.25	2.27
	XP_721489.2	HOL1	Multidrug resistance protein: MFS transporter	586	0.92	1.72
	XP_712971.2	HOL4	Multidrug resistance protein: ion transporter	624	−0.5	−0.73
	XP_714342.2	QDR3	Multidrug resistance protein: membrane transporter	697	−0.59	−0.95
	KGU11486.1	CDR1	Multidrug resistance protein	1501	−1.65	−2.4
Chitin degradation	XP_718674.1	CHT1	Putative Zn(II)2Cys6 transcription factor	1389	1.14	1.33
	XP_721807.2	CHT2	GPI-linked chitinase	1752	−0.51	−1.33
	XP_722560.1	CHT4	Chitinase	1167	2.37	2.62

**Table 2 molecules-24-01587-t002:** ^13^C NMR data of compound **1** and sasanquasaponin III (C_5_D_5_N, ppm).

No.	1	Sasanquasaponin III ^a^	No.	1	Sasanquasaponin III ^a^
1	38.8	38.8	22-*O*-Ang 1‴″	167.8	167.7
2	26.3	26.3	2‴″	129.2	129.7
3	89.9	89.6	3‴″	136.5	136.3
4	39.5	39.5	4‴″	15.7	15.7
5	55.3	55.4	5‴″	20.7	20.8
6	18.5	18.7	GlcA 1′	105.2	105.3
7	36.7	36.6	2′	79.2	79.3
8	41.5	41.6	3′	81.8	82.5
9	46.9	47.0	4′	71.1	71.1
10	36.7	36.8	5′	76.8	76.9
11	23.7	23.8	6′	172.3	172.3
12	124.6	124.7	Glc 1″	102.1	102.5
13	144.2	144.3	2″	76.8	76.3
14	47.5	47.6	3″	78.0	78.3
15	67.4	67.4	4″	72.5	72.4
16	74.8	74.9	5″	78.0	78.1
17	45.0	45.1	6″	63.4	63.5
18	41.5	41.6	Gal 1‴	100.7	101.2
19	46.9	46.9	2‴	76.1	76.1
20	31.8	31.9	3‴	75.9	75.9
21	41.2	41.4	4‴	71.1	71.2
22	72.4	72.6	5‴	78.0	77.3
23	27.6	27.8	6‴	62.2	61.8
24	16.5	16.6	Rha 1″″	102.3	102.2
25	15.7	15.6	2″″	72.7	72.5
26	17.4	17.4	3″″	72.4	72.5
27	21.2	21.2	4″″	73.8	73.7
28	62.7	62.8	5″″	69.6	69.7
29	33.3	33.4	6″″	17.9	18.1
30	25.0	25.1			

^a^ The ^13^C-NMR data of sasanquasaponin III was referenced to the study of Matsuda et al. [21].

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
