# Peer review of "Anticandidal Potential of Stem Bark Extract from Schima superba and the Identification of Its Major Anticandidal Compound"

_molecules, 2019, doi:10.3390/molecules24081587_

Round 1
Reviewer 1 Report
Page 2, line 39: This sentence does not make sense; what do you mean by “but also the positive control AA (purity >99%)” ?
For Figure 1, it would be useful to include additional concentrations of SSE between 0 and 0.0625 in the right panel (ABTS) so that there is actually a dose response curve because the first concentration tested gives such a high %. This also applies to the left panel (DPPH) where an additional concentration of SSE between 0.0625 and 0.125 in the left panel (DPPH) would give a smoother, more apparent dose-response curve.
The data referred to on page 3, line 8, for disc diffusion assays with bacteria should be shown, either in the main text or in the supplement. Seeing these data is of additional interest because the antimicrobial activity of saponins is referenced in line 12 on page 4, so one would generally expect activity against bacteria here.
In both Figure 2 and 7, clearing is visible at the 0.0625 concentration, not just once you get to at least 0.125.
Figure 3: Include more concentrations between 0.25 and 0.5 to generate an actual curve rather than just a line indicating almost no response to maximal response.
Figure 4 is not legible.
Page 5, line 25: How were the 8 genes selected?
Page 5, lines 26-27: Please explain how the qPCR and RNA-seq results are consistent. For example, referring to ASR1, it is not apparent how a log2 ratio of -1.42 at 0.2 mg/mL going to -2.0 at 0.4 mg/mL is equivalent to a fold-change of -4.58 at 0.2 mg/mL going to -20.56 at 0.4 mg/mL. For the most part, it seems that qPCR indicates much larger changes in gene expression; and for HOL1, qPCR indicates a decrease in sensitivity upon increasing [SSE] treatment, while RNA-seq suggests the opposite.
Table 1 should not be split over 2 pages. It would also be helpful if Table 2 were not split over 2 pages.
Figure 5: When describing the data presented in this figure, the authors sometimes describe the SSE-treated cells as being “twisted” which is very misleading. The cells simply appear deflated or wrinkled, likely as a result of lysis, not “twisted”.
Figure 6: Numerical axis labels are not legible.
Figure 6: What is the minor peak in B that appears shortly after the main sasanquasaponin III peak?
Page 9, line 2-3: How pure was the sasanquasaponin III after preparation? Was the minor peak seen in Figure 6, part B, still present? If the minor peak seen in Figure 6, part B, is still present, how do you know that that compound, rather than sasanquasaponin III, is not responsible for or contributing to the anticandidal activity?
Figure 8: Either describe in the Methods or the figure caption how the data analysis was done. I assume it was a ratio of OD readings, with 0 being maximum?
Page 10, line 11: I do not believe any of the active ingredients referenced here are actually on the market as anticancer drugs, thus it is inappropriate to refer to them as “pharmaceutical agents”. Pharmaceutical agents refers to drugs that are actually prescribed and used in humans, not lead compounds or compounds in development for eventual human use.
Page 11, Lines 32-34: Rephrase this sentence. It does not seem to make sense as it is currently written.
Based on the information provided in the last paragraph of the Discussion (page 11, Lines 32-48), it is not clear why the authors focused so much effort on SSE, particularly in the RNA-seq and qPCR experiments, rather than using sasanquasaponin III for these experiments. It seems the authors try to attribute most of the anticandidal activity of SSE to this compound, but only specifically show that sasanquasaponin III can inhibit C. albicans growth and do not show whether it is responsible for the other changes reported as a result of SSE treatment.
Author Response
We would like to thank the reviewers for the constructive comments regarding the current manuscript. In response to these comments, we now have updated the manuscript with some modification. Please note that all changes in the manuscript revision are highlighted in yellow.
Reviewer 1
1. Page 2, line 39: This sentence does not make sense; what do you mean by “but also the positive control AA (purity >99%)” ?
Ans. We apologize to our unclear description. The sentence has been modified. SSE scavenged DPPH and ABTS+ radical in a dose-dependent manner as well as the positive control of ascorbic acid (purity > 99%).
2. For Figure 1, it would be useful to include additional concentrations of SSE between 0 and 0.0625 in the right panel (ABTS) so that there is actually a dose response curve because the first concentration tested gives such a high %. This also applies to the left panel (DPPH) where an additional concentration of SSE between 0.0625 and 0.125 in the left panel (DPPH) would give a smoother, more apparent dose-response curve.
Ans. In this study, we want to investigate if the S. superba extract had any biological function, so several tests including antioxidant and antimicrobial activities were extensively determined. Because the antioxidants were not particularly prominent, the further analysis of antioxidants was not necessary. This study focused on the anticandidal characterization of S. superba extract. Consequently, sasanquasaponin III purified from S. superba extract was identified as its anticandidal ability.
3. The data referred to on page 3, line 8, for disc diffusion assays with bacteria should be shown, either in the main text or in the supplement. Seeing these data is of additional interest because the antimicrobial activity of saponins is referenced in line 12 on page 4, so one would generally expect activity against bacteria here.
Ans. No clear zone was observed in Escherichia coli, Klebsiella oxytoca, and Staphylococcus aureus tests. These results were negative and had been described in the Results and Discussion of the manuscript. Page 4, line 12, indicated that saponins can perform antimicrobial activity [18]. The saponin derived from the pulp of Sapindus mukorossi was also containing antifungal activity, and not antibacterial activity. To avoid misunderstanding, the sentence has been modified according to reviewer 2’s suggestion.
4. In both Figure 2 and 7, clearing is visible at the 0.0625 concentration, not just once you get to at least 0.125.
Ans. We thank the reviewer for the positive comments on our manuscript. The sentences have been modified.
5. Page 5, line 25: How were the 8 genes selected?
Ans. The eight genes which had been annotated as known genes and comprised of four upregulated and four downregulated genes were randomly selected to confirm the RNA-seq results. The sentences of real-time RT-PCR have been modified in Materials and methods.
6. Page 5, lines 26-27: Please explain how the qPCR and RNA-seq results are consistent. For example, referring to ASR1, it is not apparent how a log2 ratio of -1.42 at 0.2 mg/mL going to -2.0 at 0.4 mg/mL is equivalent to a fold-change of -4.58 at 0.2 mg/mL going to -20.56 at 0.4 mg/mL. For the most part, it seems that qPCR indicates much larger changes in gene expression; and for HOL1, qPCR indicates a decrease in sensitivity upon increasing [SSE] treatment, while RNA-seq suggests the opposite.
Ans. The two technologies, qPCR and RNA-seq, are different, and the amount of reaction used in this study was also different. Therefore, the fold change was not the same. Nevertheless, the all trend of qPCR results was consistent with the RNA-seq result. The similar result can be observed in Yang et al. study. Although the fold change of HOL1 expression in qPCR result was not increased along with the addition of S. superba extract, the up-regulation of HOL1 expression can be observed as the treatment of 0.2 mg/mL and 0.4 mg/mL S. superba extract. Taking together, the gene expression regulated by S. superba extract was reliable in the RNA-seq result.
Reference: Yang, Q.; Gao, L.; Tao, M.; Chen, Z.; Yang, X.; Cao, Y., Transcriptomics analysis of Candida albicans treated with Huanglian Jiedu decoction using RNA-seq. Evid. Based Complement. Alternat. Med. 2016, 2016, 3198249.
Supporting Information TABLE S2: A list of primers used in real-time PCR analysis and comparison in the changes of gene expression determined by RNA-Seq and real-time PCR approaches. | |||
Target gene | Primer sequence (5′-3′) | log2Ratio determined by RNA-Seq | Fold change of the gene expression determined by real-time PCR |
Actin | Actin-F:CCAGAAGCTTTGTTCAGACCA | ||
Actin-R:AGAACCACCAATCCAGACAGA | |||
SPO22 | SPO22-F: AGGAAATGAAAAATTGGCACT | 7.7 | 4.2 |
ERG9-R: AAAATCTTGACAACCCCTGG | |||
ERG11 | ERG11-F: CCCTGAAGATTTTGATCCAAC | 2 | 2.8 |
ERG11-R: GGTTCCCAATTGAACATAAGC | |||
ERG24 | ERG24-F: ATGCTGGCTGGCCTATTTA | 2.5 | 1.5 |
ERG24-R: TTCAAGAGAGCTGTCGCTGTT | |||
CDR1 | CDR1-F: CCAGGATTTTGGATTTTCATG | 2.4 | 4.1 |
CDR1-R: CAGAATGCACAAGATCCATCA | |||
CDR2 | CDR2-F: AATGCTGCTAATTTGGCTACA | 5.3 | 3.6 |
CDR2-R: AAATAACCACCAGCAGCTTCG | |||
CDR4 | CDR4-F: TGACAAAGGAGCAGTTACCAG | -2.5 | 0.02 |
CDR4-R: CCTGCAACCTTCATATATGGC | |||
Ndt80 | Ndt80-F: CCTGACCACGAGACTGTCAAA | 4.9 | 1.8 |
Ndt80-R: CCCAATAATTCGACATGCAAG | |||
FLU1 | FLU1-F: TGGCTTGGTTGGACTGGTAAT | 1.8 | 2.1 |
FLU1-R: CCCTAAAAGTGTACTTGCCCA | |||
7. Table 1 should not be split over 2 pages. It would also be helpful if Table 2 were not split over 2 pages.
Ans. We thank the reviewer for the positive comments on our manuscript. We will provide the appropriate arrangement of Figures and Tables to the Journal.
8. Figure 5: When describing the data presented in this figure, the authors sometimes describe the SSE-treated cells as being “twisted” which is very misleading. The cells simply appear deflated or wrinkled, likely as a result of lysis, not “twisted”.
Ans. We thank the reviewer for the positive comments on our manuscript. The sentence has been modified as wrinkled morphology.
9. Figure 6: Numerical axis labels are not legible.
Ans. The resolution of Fig. 6 has been improved.
10. Figure 6: What is the minor peak in B that appears shortly after the main sasanquasaponin III peak?
Ans. Sasanquasaponin III and SSE were dissolved in methanol (AR). According to the UPLC profile of the control without the SSE (Fig. 6C), the minor peak may be the impurity in the mobile phase including acetonitrile and 0.1% trifluoroacetic acid water. The concentration of impurity was low and did not affect our analysis.
11. Page 9, line 2-3: How pure was the sasanquasaponin III after preparation? Was the minor peak seen in Figure 6, part B, still present? If the minor peak seen in Figure 6, part B, is still present, how do you know that that compound, rather than sasanquasaponin III, is not responsible for or contributing to the anticandidal activity?
Ans. Based on the blank contrail, the minor peak was present in the mobile phase (acetonitrile and 0.1% trifluoroacetic acid water), and not in sasanquasaponin III. In addition, the result of TLC and NMR revealed that no other impurity existed in our sasanquasaponin III sample. In this study, acetonitrile was purchased from TEDIA (Tedia company, USA). Trifluoroacetic acid was purchased from MACKLIN (Shanghai Macklin Biochemical company, China). Methanol was purchased from XiLONG SCIENTIFIC (Xilong Scientific company, China).
12. Figure 8: Either describe in the Methods or the figure caption how the data analysis was done. I assume it was a ratio of OD readings, with 0 being maximum?
Ans. We thank the reviewer for the positive comments on our manuscript. The calculation was added into the Materials and methods of the main text.
Anticandidal activity analysis was calculated as follows:
Anticandidal activity (%)=[absorption of untreated sample (0 mg/mL) –absorption of treated sample (0.03125~1 mg/mL)]/absorption of untreated sample (0 mg/mL)×100.
13. Page 10, line 11: I do not believe any of the active ingredients referenced here are actually on the market as anticancer drugs, thus it is inappropriate to refer to them as “pharmaceutical agents”. Pharmaceutical agents refers to drugs that are actually prescribed and used in humans, not lead compounds or compounds in development for eventual human use.
Ans. We thank the reviewer for the positive comments on our manuscript. The sentence has been modified. S. superba is widely distributed in southeast China and composed of bioactive ingredients which can be applied in cancer cell lines.
14. Page 11, Lines 32-34: Rephrase this sentence. It does not seem to make sense as it is currently written.
Ans. We thank the reviewer for the positive comments on our manuscript. The sentence has been modified. Saponins have a cardioprotective potential because of their antihypoxic, calcium ion regulation/inotropic, anti-atherosclerotic, and hypolipidemic effects, anoxia/reoxygenation, cardiocyte apoptosis, and vasodilatory/cardiac depressant [29]. Additionally, saponins from plants exhibit anticandidal susceptibilities, such as anagallisin C, hostaside I, hostaside II, and CAY1, provide a promising evidence for antifungal ability [14, 17, 30, 31].
15. Based on the information provided in the last paragraph of the Discussion (page 11, Lines 32-48), it is not clear why the authors focused so much effort on SSE, particularly in the RNA-seq and qPCR experiments, rather than using sasanquasaponin III for these experiments. It seems the authors try to attribute most of the anticandidal activity of SSE to this compound, but only specifically show that sasanquasaponin III can inhibit C. albicans growth and do not show whether it is responsible for the other changes reported as a result of SSE treatment.
Ans. Schima superba was a traditional Chinese medicine used for heat clearing, detoxicating, and treating furuncle. These herbs were usually treated by boiling to obtain the bioactive extract. Therefore, according to our results, the anticandidal property can be used as another function for Schima superba in complementary and alternative medicine. Additionally, based on the result of UPLC analysis equipped with a Waters ACQUITY UPLC HSS T3 column (2.1 mm × 100 mm, 1.8 μm) and UV detector, one major peak was observed. Hence, it was reasonable to speculate its anticandidal property. An evident clear zone and C. albicans inhibition were also determined by adding different sasanquasaponin III concentrations. As for the other functions and activities of sasanquasaponin III, further study is necessary.
Reviewer 2 Report
The manuscript entitled “Anticandidal potential of stem bark extract from Schima superba and the identification of its major anticandidal compound” authored by Chun Wu et al. shows a characterization of the substances present in the Schima superba extract followed by assays to determine the extract main componentes, its antimicrobial activity as well as the determination of its molecular targets. The authors use the advanced sophisticated testing equipments in their experiments.
I make some suggestions in order to contribute to the improvement of the manuscript:
1-I suggest that authors seek the assistance of an English-speaking native or a specialized editing service to review the manuscript. Although the article does not present grammatical errors, several style errors are described. For example, "a clear zone against C. albicans " should be substituted by “a clear inhibition zone was observed for C. albicans growth" or “SSE significantly prohibited C. albicans growth", changed to “SSE significantly inhibited C. albicans growth” and others.
2- The abstract should be reorganized and a conclusion should be added.
3- P.2, lines 16-17: “ergosterol, which is a major constituent of the fungal membrane”. Please, change to ergosterol, “which is a major lipid component of fungal ..”
4- P.2, line 38: “SSE had a significant antioxidant activity (Figure 1).” The authors do not mention the use of statistical methods when affirming “significant” antioxidant activity.
5- P.2, line 39: “SSE scavenged not only DPPH and ABTS + radical in a dose-dependent manner but also the positive control AA (purity>99%). Please explain better what you meant by this sentence.
6- P.3, lines 12-13: The authors used "half maximal inhibitory concentration "which corresponds to IC 50 as a synonym of MIC 50. These are different ways of determining the inhibition of microbial growth. Please see: P Van Dijck. Microbial Cell, Vol. 5, No. 7, pp. 300 - 326; doi: 10.15698/mic2018.07.638
7- P.4 , lines 9-12: Saponins have a cardioprotective potential because of their antihypoxic, calcium ion regulation/inotropic, anti-atherosclerotic, and hypolipidemic effects, anoxia/reoxygenation, cardiocyte apoptosis, and vasodilatory/cardiac depressant [17]. Additionally, saponins can perform antimicrobial activity [18]. Please move the sentences for introduction or discussion section.
8- P.4, lines 19-20: “Pearson correlation coefficients were determined to reflect the gene expression correlation...” I suggest that the authors describe which statistical methods were used for all comparisons made in the study.
9- P.5. Figure 4. All figures should be sent in separate files to increase resolution and allow to analysis by reviewers. In the current Figure 4 version, there is no possibility of analysis.
10- Table S2 - I suggest that preferably all hypothetical proteins be withdrawn or that by means of comparative similarity analyzes the possible protein is indicated (CLUSTAL Multiple Sequence Alignment or similar).
11- P.5, lines 17-18: “Therefore, the effects of SSE on the genes involved in ergosterol biosynthesis pathway and multidrug transporter are shown in Table 1” Authors should review this statement. The genes ERG1, ERG7, ERG9, ERG25, ERG26, and ERG27 have been proved to be essential in ergosterol biosynthesis which are not mentioned in Table 1 (Q Lv et al., 2016). The title of the Tables (1, and S1, S2) describe down-regulated and up-regulated genes. Were they genes or proteins? Please review.
(Q Lv et al. The synthesis, regulation, and functions of sterols in Candida albicans: Well-known but still lots to learn. VIRULENCE, 7 (6): 649–659, 2016).
13- P. 12, 4.1. Plant material and extraction. I suggest that the authors describe the exact location (latitude and longitude) of the plant material collection. Plants chemical content varies depending on the soil quality, temperature, rainfall among others.
14- P.12, 4.4. Antimicrobial assay: I suggest that the authors review the methodologies used (disc-diffusion and microdilution in broth). The CLSI and EUCAST have standardized the methodology for the culture medium (not mentioned), inoculum size (not described), drug and control strains (not described), incubation time (not described), breakpoints values (not described) and interpretation criteria (not described), controls (medium and organism growth).
15- P.13, 4.6. RNA-seq analysis of C. albicans. Please describe the methodology used for RNA quantification and evaluation of its integrity.
16- P.13., line 17: “ RSEM, cluster, DEGseq, GO annotation, KEGG annotation, and BLAST for genome mapping, ....” Please, write the meaning of each acronym and the electronic address of each consulted database.
17- P.13, 4.7. Quantitation of gene expression by real-time RT-PCR. I suggest that the authors use a table to describe the oligonucleotides used. The amplification solution constitution (number of primers, amount of DNA, amount of buffer, ...) must also be described.
Author Response
We would like to thank the reviewers for the constructive comments regarding the current manuscript. In response to these comments, we now have updated the manuscript with some modification. Please note that all changes in the manuscript revision are highlighted in yellow.
Reviewer 2
1. I suggest that authors seek the assistance of an English-speaking native or a specialized editing service to review the manuscript. Although the article does not present grammatical errors, several style errors are described. For example, "a clear zone against C. albicans " should be substituted by “a clear inhibition zone was observed for C. albicans growth" or “SSE significantly prohibited C. albicans growth", changed to “SSE significantly inhibited C. albicans growth” and others.
Ans. We thank the reviewer for the positive comments on our manuscript. The manuscript has been checked by language editing services before submission. We had edited the manuscript again.
2. The abstract should be reorganized and a conclusion should be added.
Ans. The abstract has been modified, and the conclusion has been added.
3. P.2, lines 16-17: “ergosterol, which is a major constituent of the fungal membrane”. Please, change to ergosterol, “which is a major lipid component of fungal.
Ans. We thank the reviewer for the positive comments on our manuscript. The sentence has been modified according to the reviewer’s suggestion.
4. P.2, line 38: “SSE had a significant antioxidant activity (Figure 1).” The authors do not mention the use of statistical methods when affirming “significant” antioxidant activity.
Ans. We thank the reviewer for the positive comments on our manuscript. The sentence has been modified. SSE had a high antioxidant activity (Figure 1).
5. P.2, line 39: “SSE scavenged not only DPPH and ABTS + radical in a dose-dependent manner but also the positive control AA (purity>99%). Please explain better what you meant by this sentence.
Ans. We apologize to our unclear description. The sentence has been modified. SSE scavenged DPPH and ABTS+ radical in a dose-dependent manner as well as the positive control of ascorbic acid (purity > 99%).
6. P.3, lines 12-13: The authors used "half maximal inhibitory concentration "which corresponds to IC 50 as a synonym of MIC 50. These are different ways of determining the inhibition of microbial growth. Please see: P Van Dijck. Microbial Cell, Vol. 5, No. 7, pp. 300 - 326; doi: 10.15698/mic2018.07.638
Ans. We apologize to our inappropriate description and thank the reviewer for the positive comments on our manuscript. Our results should be indicated as IC50. These sentences have been modified.
7. P.4 , lines 9-12: Saponins have a cardioprotective potential because of their antihypoxic, calcium ion regulation/inotropic, anti-atherosclerotic, and hypolipidemic effects, anoxia/reoxygenation, cardiocyte apoptosis, and vasodilatory/cardiac depressant [17]. Additionally, saponins can perform antimicrobial activity [18]. Please move the sentences for introduction or discussion section.
Ans. We thank the reviewer for the positive comments on our manuscript. These sentences have been moved to Discussion. Saponins have a cardioprotective potential because of their antihypoxic, calcium ion regulation/inotropic, anti-atherosclerotic, and hypolipidemic effects, anoxia/reoxygenation, cardiocyte apoptosis, and vasodilatory/cardiac depressant [29]. Additionally, saponins from plants exhibit anticandidal susceptibilities, such as anagallisin C, hostaside I, hostaside II, and CAY1, provide a promising evidence for antifungal ability [14, 17, 30, 31].
8. P.4, lines 19-20: “Pearson correlation coefficients were determined to reflect the gene expression correlation...” I suggest that the authors describe which statistical methods were used for all comparisons made in the study.
Ans. Pearson correlation coefficients were determined using cor by the R software. The method has been added into the Materials and methods.
9. P.5. Figure 4. All figures should be sent in separate files to increase resolution and allow to analysis by reviewers. In the current Figure 4 version, there is no possibility of analysis.
Ans. The high-resolution figures have been sent in separate files to the journal when we first submitted. The resolution of Fig. 4 has been improved. The detail description has been added in the figure legend.
Figure 4. The volcano plots of the differentially expressed genes of C. albicans treated with SSE at the concentrations of 0, 0.2, and 0.4 mg/mL for RNA-seq analysis. The red and blue dots indicated the significantly differentially expressed genes with up-regulation and down-regulation, respectively. The red dot indicates up-regulation (Up); the blue dot indicates down-regulation (Down); the grey dot indicates no differentially expressed genes (no-DEGs).
10. Table S2 - I suggest that preferably all hypothetical proteins be withdrawn or that by means of comparative similarity analyzes the possible protein is indicated (CLUSTAL Multiple Sequence Alignment or similar).
Ans. These hypothetical proteins in Table S1 and S2 have been annotated.
11. P.5, lines 17-18: “Therefore, the effects of SSE on the genes involved in ergosterol biosynthesis pathway and multidrug transporter are shown in Table 1” Authors should review this statement. The genes ERG1, ERG7, ERG9, ERG25, ERG26, and ERG27 have been proved to be essential in ergosterol biosynthesis which are not mentioned in Table 1 (Q Lv et al., 2016). The title of the Tables (1, and S1, S2) describe down-regulated and up-regulated genes. Were they genes or proteins? Please review.
Ans. We thank the reviewer for the positive comments on our manuscript. In ergosterol biosynthesis pathway, two genes encoding C-5 sterol desaturase and methylsterol monooxygenase with over a twofold change were significantly downregulated in a dose-dependent manner. The other ergosterol biosynthesis genes were not markedly downregulated; therefore, not all ergosterol biosynthesis genes were listed in Table 1. According to the reviewer’s suggestion, the other genes involved in ergosterol biosynthesis pathway were added into Table 1.
12. P. 12, 4.1. Plant material and extraction. I suggest that the authors describe the exact location (latitude and longitude) of the plant material collection. Plants chemical content varies depending on the soil quality, temperature, rainfall among others.
Ans. The exact location has been provided according to your suggestion in our manuscript. The stem bark of S. superba was collected from Sanming (longitude: 116.820837, latitude: 26.972907), Fujian, China.
13. P.12, 4.4. Antimicrobial assay: I suggest that the authors review the methodologies used (disc-diffusion and microdilution in broth). The CLSI and EUCAST have standardized the methodology for the culture medium (not mentioned), inoculum size (not described), drug and control strains (not described), incubation time (not described), breakpoints values (not described) and interpretation criteria (not described), controls (medium and organism growth).
Ans. We thank the reviewer for the positive comments on our manuscript. The antimicrobial assay used in this study has been described in the Materials and methods.
14. P.13, 4.6. RNA-seq analysis of C. albicans. Please describe the methodology used for RNA quantification and evaluation of its integrity.
Ans. The NanoDrop 2000 (Thermo Fisher Scientific, Wilmington, DE, USA) was used for RNA quantification. The sentence has been added in the Materials and methods.
15. P.13., line 17: “ RSEM, cluster, DEGseq, GO annotation, KEGG annotation, and BLAST for genome mapping, ....” Please, write the meaning of each acronym and the electronic address of each consulted database.
Ans. The sentences and references have been modified and cited in the Materials and methods, respectively.
(1) Kim, D., Langmead, B. & Salzberg, S. L. HISAT: a fast spliced aligner with low memory requirements. Nat. Methods 12, 357-360 (2015).
(2) Li, B. & Dewey, C. N. RSEM: accurate transcript quantification from RNA-Seq data with or without a reference genome. BMC Bioinformatics 12, 323 (2011).
(3) Eisen, M. B., et al. Cluster analysis and display of genome-wide expression patterns. Proc Natl Acad Sci USA, (1998)95(25): 14863-8. 2001.29: 1165-1188 (2001).
(4) M. J. L. de Hoon, et al. Open Source Clustering Software. Bioinformatics, 20(9): 1453-1454 (2004).
(5) Wang, L. et al. DEGseq: an R package for identifying differentially expressed genes from RNA-seq data. Bioinformatics 26, 136-138 (2010).
(6) Ashburner M, Ball CA, Blake JA, Botstein D, Butler H, Cherry JM, Davis AP, Dolinski K, Dwight SS, Eppig JT, Harris MA, Hill DP, Issel-Tarver L, Kasarskis A, Lewis S, Matese JC, Richardson JE, Ringwald M, Rubin GM, Sherlock G. Gene ontology: tool for the unification of biology. The Gene Ontology Consortium. Nat Genet. 25:25-29 (2000)
(7) Kanehisa, M., Goto, S., Hattori, M., Aoki-Kinoshita, K. F., Itoh, M., Kawashima, S., Katayama, T., Araki, M., and Hirakawa, M. From genomics to chemical genomics: new developments in KEGG, Nucleic Acids Res., 34, D354-357 (2006).
16. P.13, 4.7. Quantitation of gene expression by real-time RT-PCR. I suggest that the authors use a table to describe the oligonucleotides used. The amplification solution constitution (number of primers, amount of DNA, amount of buffer, ...) must also be described.
Ans. The sentences have been modified, and the primer sequences were listed as a table. Real-time PCR was performed using an UltraSYBR Mixture (CWBiotech) in a final volume of 12.5 μL (2×UltraSYBR Mixture 6.25 μL, 10 μM Forward primer 0.25 μL, 10 μM Reverse primer 0.25 μL, cDNA 2 μL, and RNase-free water 3.75 μL) for each reaction in the Roche LightCycler® 480 System (Roche Group, Switzerland).
Table S4. The primer sets of C. albicans genes were used in real-time RT-PCR for quantitation of gene expression.
Gene ID | Primer sequence (5’-3’) |
18SrRNA | Forward: CGATGGAAGTTTGAGGCAAT |
Reverse: CACGACGGAGTTTCACAAGA | |
3638048 | Forward: TGTTGGTTCCTTGATTGCC |
Reverse: GGCTCTACCTGCAATAAGCAAT | |
3636838 | Forward: CAGCAGCTACTTCAAATGACG |
Reverse: CCCCATCCAATACCCAAA | |
3646427 | Forward: AAAGCATATCTTAGCCGCAG |
Reverse: GCCAATCCTTGTTCAACAAC | |
3640751 | Forward: GAATACTGTCAATCAGGTGCTG |
Reverse: GGATTCTGGCAGCTTGAAC | |
3639294 | Forward: AAAGGCCTTGTACAGACAGCT |
Reverse: TCTTGCTCGGCTTTACCA | |
3644402 | Forward: CGGATCATCAAACACTGCTAGT |
Reverse: CCAGATCCATAACTTCCACTTG | |
3635222 | Forward: CAAAGCAAGAAGAGAGTACCCC |
Reverse: CCTTGACTGGTTCATCTGGAAT | |
3640144 | Forward: GGCATATGGATGACAAAGGG |
Reverse: CGTATTCTGGTTCAGCCTTGA |

Reviewer 3 Report
The reviewed study is a fine written piece of work with well-developed research methodology. It revolves around an original subject that focuses on antimicrobial activity of an extract from Schima superba. Still, there is room for slight improvement in some aspects.
Firstly, the introduction is well-written in that it catches relevant aspects from previous works related to saponins from S. superba and its general effects on human health. Considering the following analysis of polyphenols using Folin-Ciocalteu assay, it would be advisable to have a very brief overview based on proper literature focusing on relevant polyphenolic content and activity assays performed in similar plants to have a general idea of what to expect. All information related to Candida albicans is relevant for understanding the following methodology.
Secondly, the results section contains proper data and images selection. The MIC50 value determined in section 2.2 should be presented along with standard deviation. However, when it comes to polyphenol and saponin content determination (section 2.3), it would be recommended to include the value of R-squared for each standard curve (for both gallic acid and oleanic acid) in order to prove the linearity of content determination. Also, determined values of gallic acid and oleanic acid for the studied extract should be presented with the calculated standard deviation for proper statistical reliability. Considering the importance of the determined compound, sasanquasaponin III, its structure should be shown along with NMR results in the paper, not just in the supplementary material (section 2.6).
Thirdly, the discussion part undelines and correlates the results very well. The attention to details is shown by the overview of biochemical processes undelying the antifungal activity. A small mistake appears at page 11, line 23- the species name should be written in lower case. Also, it would be interesting if the authors would make a correlation between DPPH, ABTS+ and polyphenolic content and to emphasize the antioxidant activity of this extract. Also, the oleanic acid content should be presented with the determined standard deviation.
Finally, the research methodology developed in the materials and methods section is well-suited for the study and it is according to previously described assays from literature. As a small suggestion, the extraction of saponins would work better by use of a non-polar extraction solvent.
In conclusion, I believe this article presents an original piece of work that has a potential soundness to readers interested in the topic. With slight touches this manuscript can be improved for the best.
Author Response
We would like to thank the reviewers for the constructive comments regarding the current manuscript. In response to these comments, we now have updated the manuscript with some modification. Please note that all changes in the manuscript revision are highlighted in yellow.
Reviewer 3
1. Firstly, the introduction is well-written in that it catches relevant aspects from previous works related to saponins from S. superba and its general effects on human health. Considering the following analysis of polyphenols using Folin-Ciocalteu assay, it would be advisable to have a very brief overview based on proper literature focusing on relevant polyphenolic content and activity assays performed in similar plants to have a general idea of what to expect. All information related to Candida albicans is relevant for understanding the following methodology.
Ans. We thank the reviewer for the positive comments on our manuscript. The sentences have been added in the Introduction.
Bioactive compounds from S. superba have been widely explored. An abundance of phenolics which had antimicrobial and antioxidant potential was investigated in S. superba [3, 4]. Saponins from S. superba leaves have antifungal effects against Magnaporthe oryzae, which can cause a highly damaging disease in rice [5, 6].
2. Secondly, the results section contains proper data and images selection. The MIC50 value determined in section 2.2 should be presented along with standard deviation. However, when it comes to polyphenol and saponin content determination (section 2.3), it would be recommended to include the value of R-squared for each standard curve (for both gallic acid and oleanic acid) in order to prove the linearity of content determination. Also, determined values of gallic acid and oleanic acid for the studied extract should be presented with the calculated standard deviation for proper statistical reliability. Considering the importance of the determined compound, sasanquasaponin III, its structure should be shown along with NMR results in the paper, not just in the supplementary material (section 2.6).
Ans. We thank the reviewer for the positive comments on our manuscript. The R-squared for each standard curve (for both gallic acid and oleanic acid) has been added in the manuscript. y = 0.0296x - 0.0085 (R2 = 0.977) for gallic acid; y = 0.5956x + 0.1045 (R2 = 0.9987) for oleanic acid. The structure of sasanquasaponin III has been added in the manuscript as Fig. 7.
3. Thirdly, the discussion part undelines and correlates the results very well. The attention to details is shown by the overview of biochemical processes undelying the antifungal activity. A small mistake appears at page 11, line 23- the species name should be written in lower case. Also, it would be interesting if the authors would make a correlation between DPPH, ABTS+ and polyphenolic content and to emphasize the antioxidant activity of this extract. Also, the oleanic acid content should be presented with the determined standard deviation.
Ans. We thank the reviewer for the positive comments on our manuscript. Chinese herb medicine is named according to its genus name and medicinal parts. Therefore, both words should be capitalized. A correlation between DPPH, ABTS+ and polyphenolic content has been described in the Discussion. The standard deviation of polyphenolic and saponin content corresponding to gallic acid and oleanic acid has been added in the manuscript. Polyphenolic content: 256.6±5.1 µg/mg; Saponin content: 357.8±31.5 µg/mg.
4. Finally, the research methodology developed in the materials and methods section is well-suited for the study and it is according to previously described assays from literature. As a small suggestion, the extraction of saponins would work better by use of a non-polar extraction solvent.
Ans. We thank the reviewer for the positive comments on our manuscript. At the beginning of this study, the anticandidal effect of saponins was not understood in the S. superba extract. Therefore, the total extract of polyphenols and saponins was conducted by using alcohol (Yoshikawa et al., 2000; Awaad et al., 2006).
(1) Yoshikawa, K., Hirai, H., Tanaka, M., Arihara, S. Antisweet natural products. XV. Structures of Jegosaponins A-D from Styrax japonica Sieb. et Zucc. Chem Pharm Bull (Tokyo). 48, 1093-1096 (2000).
(2) Awaad, A.S., Maitland, D.J., Soliman, G.A. Hepatoprotective activity of Schouwia thebica webb. Bioorg Med Chem Lett. 16, 4624-4628 (2006).
Round 2
Reviewer 2 Report
The manuscript revised version presents more clarity and detail at many points in the manuscript. However, I believe that the authors did not understand the correction request in Table 1. "Response to SSE of the genes involved in ergosterol biosynthesis, ...." The authors present in this table several proteins: lanosterol synthase (gene encoding ERG7); 3-keto-steroid reductase (gene encoding ERG27); bifunctional farnesyl-diphosphate farnesyltransferase / squalene synthase (gene encoding ERG9?); phosphomevalonate kinase (gene encoding ERG8); methylsterol monooxygenase (gene encoding ERG25) which are encoded by different ERG genes. I advise the authors to put in the table a column for the gene encoding each protein described or change the title of the table.The XP_ codes correspond to the proteins entries.
2- What are the genes corresponding to the fragments listed in Table S4 in the reply letter? Please correct Table S4.
3- The authors also did not understand what is IC50 As the inhibition was calculated according to the formula presented, I suggest that the authors cite only that the compound inhibited by 50% the C. albicans growth when exposed to the concentration of ....
4- The antimicrobial activity assays have no validation. It was not described in both methodologies (disc-diffusion and microdilution in broth) the culture medium used, the size of the inoculum, the control drug, the control strains. The equation presented in this version of the manuscript was not referenced or demonstrated as the authors developed it. Was the compound solubilized in some solvent (concentration)? Was it added as a powder to the culture medium?
Author Response
The detailed response with Figures was uploaded as the attached file.
We would like to thank the reviewer for the constructive comments regarding the current manuscript. In response to these comments, we now have updated the manuscript with some modification. Please note that all changes in the manuscript revision are highlighted in yellow.
1. The manuscript revised version presents more clarity and detail at many points in the manuscript. However, I believe that the authors did not understand the correction request in Table 1. "Response to SSE of the genes involved in ergosterol biosynthesis, ...." The authors present in this table several proteins: lanosterol synthase (gene encoding ERG7); 3-keto-steroid reductase (gene encoding ERG27); bifunctional farnesyl-diphosphate farnesyltransferase / squalene synthase (gene encoding ERG9?); phosphomevalonate kinase (gene encoding ERG8); methylsterol monooxygenase (gene encoding ERG25) which are encoded by different ERG genes. I advise the authors to put in the table a column for the gene encoding each protein described or change the title of the table. The XP_ codes correspond to the proteins entries.
Ans. The protein has been added in the column of Table 1.
2. What are the genes corresponding to the fragments listed in Table S4 in the reply letter? Please correct Table S4.
Ans. The protein has been added in the column of Table S4.
3. The authors also did not understand what is IC50 As the inhibition was calculated according to the formula presented, I suggest that the authors cite only that the compound inhibited by 50% the C. albicans growth when exposed to the concentration of ....
Ans. We apologize to our misunderstanding. As the reviewer’s suggestion, we had corrected the inhibition as the 50% inhibition of C. albicans.
4. The antimicrobial activity assays have no validation. It was not described in both methodologies (disc-diffusion and microdilution in broth) the culture medium used, the size of the inoculum, the control drug, the control strains. The equation presented in this version of the manuscript was not referenced or demonstrated as the authors developed it. Was the compound solubilized in some solvent (concentration)? Was it added as a powder to the culture medium?
Ans. The detailed methods have been modified in the manuscript. As the reviewer’s suggestion, the inhibition rate was shown and the equation has been deleted to avoid the misunderstanding.
A pathogen antagonistic dosage assay was conducted through the disc diffusion method [34]. C. albicans (ATCC10231) was cultivated in Sabouraud dextrose broth at 37 °C for 2 days. The culture was adjusted to obtain a concentration of approximately 108 CFU/mL. One hundred microliters of culture suspension was placed on the petri dishes of Sabouraud dextrose agar. Then, 6 mm paper discs impregnated with 20 µL of the SSE and sasanquasaponin III at the concentration of 100, 50, 25, 12.5, 6.25, 3.125, 1.5625, and 0 mg/mL dissolved in distilled water to obtain the concentration of 2, 1, 0.5, 0.25, 0.125, 0.0625, 0.03125, and 0 mg/disc. Different concentrations of SSE and sasanquasaponin III from 0 mg to 2 mg were arranged on the plate with C. albicans. The distance between discs was 2 cm. The antibiotic disc (Amphotericin B, ROSCO, Taastrup, Denmark) was used as the positive control. The plates were cultivated at 37 °C, and the clear zone of growth inhibitions was observed. The microbroth dilution was performed in 96-well plate supplemented with Sabouraud dextrose broth [35]. Different concentrations of SSE and sasanquasaponin III (0, 0.0625, 0.125, 0.25, 0.5, 1, and 2 mg/mL) were obtained by a twofold serial dilution in the Sabouraud dextrose broth at a final volume of 50 µL. The liquid culture of C. albicans was adjusted with OD600 of approximately 0.2. A microbial suspension (50 µL) was added to each well and the mixture (100 µL) was incubated at 37 °C. Afterward, the 96-well plate treated with SSE and sasanquasaponin III was analyzed and determined with a plate reader at 600 nm.
